# "Eliminating Social Distinctions" or "Preserving Social Relations": Two Explanations of *Datong* in Modern China

**Mimi Pi**

Department of Philosophy, Capital Normal University, Beijing 100089, China; 6764@cnu.edu.cn

**Abstract:** A Confucian scholar, Kang Youwei, living in the late Qing period imagined a future utopian society called *datong* which eliminated all social distinctions. To illustrate it, he borrowed and developed the theory of the Three Ages, which first appeared in the Confucian classic *Gongyang Commentary on the Spring and Autumn Annals*, as well as in the theory of human nature in the Han dynasty. However, one of his students, Chen Huanzhang, made a new explanation of the "Liyun" chapter that greatly differed from his teacher. According to him, *datong* was a society committed to "preserving social relations". The different understandings of *datong* reveals two different patterns of social relations in Confucianism. Besides the traditional *wulun* pattern, Kang Youwei offered another possible pattern. Although it proved to be a failure in practice, as a theory that discovered many hidden traditions in Confucianism, Kang Youwei's *datong* theory is worthy of attention.

**Keywords:** Liyun; *datong*; Kang Youwei; Chen Huanzhang

## 1. Introduction

The ancient Chinese society that was formed and dominated by Confucianism is generally considered to be a community based on kinship and family relations. A typical metaphor for Chinese social relations comes from the well-known sociologist Fei Xiaotong 费孝通 (1910–2005). According to him, the organizational principle of Chinese traditional society is similar to the concentric circles formed when a stone is thrown into a lake. The self is at the center, the circle immediately surrounding the self is the nuclear family (the most intimate relatives), and the outer circles resemble distant relatives and strangers. Each circle spreading out from the center becomes more distant and consequently more insignificant (Fei 2013, pp. 28–30). This means that in Chinese society the way to get along with others for an individual is not fixed, but depends on their relative position in the pattern of kinship. This is quite different from the Western scheme where all members in an organization are equivalent. The term *renlun*人伦 (human relation) is used by the Confucians to define or prescribe the relations between different kinds of people. To be specific, it includes five relations (五伦 *wulun*): those between father and son, husband and wife, elder and younger brother, ruler and ministers, and friends. In a word, the traditional Chinese society was a society formed by acquaintances. The *wulun* was the principle to deal with the relations between acquaintances.

Does this mean that according to Confucianism the *wulun* schema is the only possible pattern of social relations? No. The Confucian scholar Kang Youwei 康有为 (1858–1927) in the late Qing period put forward a new pattern which is entirely different from the *wulun* pattern[1]. By explaining the "Liyun" 礼运 (Ritual Operations) chapter of the Confucian classic the *Liji* 礼记 (Book of Rites), he imagined a future utopian society called *datong* 大同 (literally, "great unity") which eliminated all social distinctions. According to him, the *datong* society was formed by millions of completely self-sufficient, independent, and equal individuals. His work seems to show another possible way to organize the society according to Confucianism.

Kang Youwei's identity as a Confucian is a matter of doubt. A respected Confucian named Ye Dehui 叶德辉 (1864–1927) once harshly criticized him as a man "with a Confucian

appearance outside and a foreign mind inside." (Su 2002, p. 165). Apparently, Kang Youwei was viewed as a heterodox. Nonetheless, as Xiao Gongquan 萧公权 (1897–1981) has pointed out, if we do not interpret "Confucianism" in a fundamentalist way, but understand it instead as an intellectual tradition gradually formed and developed over the course of a long history, then Kang Youwei might be recognized as a revisionist of Confucianism, especially in light of his lifelong dedication to espousing and advocating for Confucianism (Xiao 2005, pp. 30–31).

With the improvement of the research on Kang Youwei, many scholars tend to agree that Kang has inherited many of the Confucian traditions. However, most of them still consider his *datong* theory inconsistent with the Confucianism because *datong* was a society where all families and social relations had vanished, which apparently contradicted to the traditional Confucian society[2]. In recent years, there are other scholars arguing that the relation between Kang Youwei's *datong* theory and the Confucian tradition should be re-explored[3].

Therefore, if we agree that Kang Youwei's *datong* theory has explored a new social pattern in Confucianism and that it needs to be explained to what extent Kang Youwei's imagined *datong* conforms to the Confucian tradition. Moreover, in order to evaluate Kang's *datong* pattern, criticism and reflection on his *datong* theory should also be taken into consideration.

## 2. Eliminating Boundaries and Distinction: Kang Youwei's Explanation of *Datong*

The term *datong* first appeared in a "Liyun" chapter of the Confucian classic the *Liji*. At the beginning, it says:

> Once upon a time, Confucius took part in a sacrifice held at the end of the year. When the ceremony was finished, he went out and climbed up the gate tower to take in the views, thereupon he let out a deep sigh. He probably sighed for the state Lu. One of his disciples called Zi You was by his side and asked, "What made you sigh, my master?" Confucius replied, "Even though I was not born during the time of the three dynasties of the Xia, Shang, and Zhou when the great way was practiced, I am still intent on seeing it realized again." Having said this, Confucius continued: "When the great way is in practice then the world as a common property (rather than that as the property of the emperor), worthy and capable people are selected for government office, and people are reliable and seek harmony amongst each other. Therefore, in such a world, people do not only show affection for their parents or for their children but they make it so that the elderly have what they need to live out their lives, that the strong are put to proper use, that the young are provided for in their growth, and that the sick, orphaned, widowed, deformed, and destitute are all taken care of. It is a time when all the men have work to do and all the women have a place to return to; a time when the people do not like to throw goods away or wish to hoard them up. It is a time when people despise not putting in their effort even it is to benefit someone else. Because of this, it is a time when schemes and intrigues are not put to use or when robbers and traitors do not exist. It is a time when it is so safe that people leave their doors open when going out. This is called *datong*." He also said: "Now the great way has disappeared and the land under heaven belongs to the royal family. Every man only loves his own parents and only cares for his own children. Goods and one's own effort are kept as one's private possessions. The sovereign passes the throne to his own son. High walls and deep moats are built to make the city safe and impregnable. Abundant thieves and robbers exist. Therefore, the rituals and righteousness are made to rule the people; the relations between ruler and ministers, father and sons, elder and younger brother, husband and wife are regulated; regulations are established and the fields are divided; the brave and the worthy are respected; people take the establishment

of merits as their own advantage, schemes, and wars arise. . . . . . . This is called *xiaokang* 小康 (the minor prosperity)". (Zheng and Kong 2008, pp. 874–76)

This passage from the "Liyun" chapter is uniquely Confucian. It gives an account of the origin and the development of *li* 礼 (variously translated into English as etiquette, ritual, rites, or ceremony). According to Confucius, *li* originated from the Three Dynasties (*sandai* 三代, that is, the Xia, Shang, and Zhou). In the Confucian tradition, the Three Dynasties were regarded as a Golden Age and resembled the highest political ideal of Confucianism. The Confucian masters Mengzi and Xunzi often started their arguments by quoting stories or sayings from the Three Dynasties. However, it is very unusual to find in the "Liyun" chapter Confucius indicating that there was a virtuous and harmonious time called *datong* that existed before the Three Dynasties. Although he did not explicitly claim that *datong* was superior to the Three Dynasties, his preference can still be inferred from the words he used, for example, the juxtaposition of *datong* and *xiaokang*.

After the failure of the political reform in 1898, Kang Youwei went into exile, travelling abroad to foreign countries. While living on Penang Island in Malaysia during the years of 1901 and 1902, he wrote a series of commentaries on several of the Confucian classics. He claimed that he had discovered an unrevealed theory of Confucius in the "Liyun." In the preface to his book *Liyunzhu* 礼运注 (Commentary on Liyun), Kang Youwei wrote:

> The way of Confucius is so magnificent. Though we cannot fully understand it, I still try my best to get a glimpse of it . . . . The moment when I started to read "Liyun", I was amazed by the great way of Confucius . . . .This is a precious book where the greatest and deepest thoughts of Confucius are preserved. It is also the best prescription to save millions of people! (Kang 2007, vol. 5, p. 553)

It is known to all that Confucius himself strictly followed regulations of rituals throughout his life. However, in the "Liyun" chapter, he exceptionally talked about a society which was not governed by rituals. For the two thousand years between Confucius and Kang Youwei, all the studies of Confucianism had focused on those visible words and sayings of Confucius. However, Kang Youwei indicated that Confucius probably kept his deepest and ingenious thoughts under cover. His few words talking about *datong* are a hint, just like the tip of an iceberg.

In his *Commentary on Liyun*, Kang Youwei offered an innovative explanation of *datong*. He explained the saying that "The world is that of the community (rather than that of the emperor), worthy and capable people are selected for government office" as "Official positions were appointed to the wise and talented who were elected by the public." He also explained the saying that "They make it so that the elderly have what they need to live out their lives, that the strong are put to proper use, that the young are provided for in their growth, and that the sick, orphaned, widowed, deformed, and destitute are all taken care of" as "They use the public property which was made up by each man's own property to provide for the aged, to care for the children, to help the poor, and to cure the sick." Again, the saying that "It is when all the men have work to do and all the women have a place to return to" was explained as "Men and women had their own authority and right which should not be exceeded by any one of them. Though not as strong as men, women were self-reliant and independent and should not be suppressed by men. Marriage for them was a contract that should be observed by both parties." (Kang 2007, vol. 5, p. 555). Generally speaking, all the governing in the time of *datong* presents a principle of equality and fairness.

Kang Youwei was not satisfied with only explaining the text of the "Liyun" and added his own interpretation: "People have this common saying, 'The empire, the state, the family.' This is a limited cognition of ancient people. Because these boundaries and distinctions of the empire, the state, the family lead to self-interest and selfishness." (Kang 2007, vol. 5, p. 555). Though the description of *datong* in the "Liyun" chapter expresses the spirit of equality, it still uses terms such as state, family, and individual; therefore, it does not present the final image of *datong* in Kang Youwei's opinion. In another book titled *Zhongyongzhu* 中庸注 (Commentary on the Zhongyong), Kang Youwei said "Confucius

had already known that there would be another sage arising three thousand years after his death. And the new sage would continue to develop his *datong* theory." (Kang 2007, vol. 5, p. 388). Therefore, for Kang Youwei, the "Liyun" chapter provided a clue to the followers to seek the realization of a real *datong*. Kang Youwei maintained that Confucius' understanding of *datong* had never been completely expressed until he had thoroughly revealed it in his *Datongshu*.

In the *Datongshu*, Kang Youwei listed all the kinds of torture and grief humans suffer in the world and attributed them to the inequality of human society. For example, as he asserted, slavery is due to class inequality, the oppression of women is due to gender inequality, and material scarcity is due to wealth inequality. After "observing all the phenomena of the world", he came to the following conclusion: "There is no way other than the way of *datong* to save humanity from suffering!" (Kang 2007, vol. 7, pp. 6–7). What exactly did the "way of *datong*" refer to? In what sense can it be called the best prescription to save humanity? According to the edition compiled by Kang Youwei's disciple Qian Dingan 钱定安, there are ten chapters in the *Datongshu*: Chapter One "Descending to the world to observe the sufferings of ordinary people", Chapter Two "Eliminating the boundaries of states and uniting the whole earth", Chapter Three "Eliminating the boundaries of nations", Chapter Four "Eliminating ethnic groups to unite all of humanity", Chapter Five "Eliminating biological difference to achieve independence", Chapter Six "Eliminating the boundary of family to become *tianmin* 天民 (people of *tian*), Chapter Seven "Eliminating category of industry to share the means of production", Chapter Eight "Eliminating struggle and disorder to realize peace and harmony", Chapter Nine "Eliminating species to love every creatures in the cosmos", and Chapter Ten "Eliminating of suffering to reach bliss." From these titles, it can be seen that Kang Youwei started his book from describing the harsh and cruel condition of human beings. After eliminating the boundaries of states, classes, nations, genders, families, and species, humanity could finally build a new social form of collective production, common distribution of resources, common living arrangements, and common welfare. In other words, the way of *datong* referred to the process called *qujie* 去界 (eliminating social boundaries and distinctions). Kang Youwei pointed out: "All that is under heaven is equal. So boundaries between states should not be built for they will lead to fighting and war. The boundaries of families should not be built for the love of humanity will be unable to spread far and wide. The boundaries of individuals should not be built for they will lead to selfishness." (Kang 2007, vol. 5, p. 555). In Kang Youwei's opinion, all the grief and sufferings originated from inequality. Inequality laid in differences. Differences led to distinction. Distinction resulted in selfishness, thus leading to differences in social relations. In turn, the inequality of society became more differences and distinction must be eliminated. It is only by and more serious. Therefore, in order to save people from sufferings and to realize the equality of the whole society, all the boundaries of groups, tribes, and communities must be eliminated, and all the selfishness and self-interest caused by doing this that the result where "everyone is unified in their equality" can be achieved[4]. (Kang 2007, vol. 5, p. 555).

### 3. The Foundation of *Datong*: "Humans Are Born by *Tian*" and the Theory of "The Three Ages"

Why did Kang Youwei consider the essence of *datong* to be where "everyone is unified in their equality"? This has to do with his theory of human nature: "humans are born by *tian*" 天. This term *tian* is semantically abundant and has been commonly translated as "nature" or "heaven" and variously refers to the sky, the natural world and its processes, and a semi-anthropomorphic deity. In his *Commentary on the Zhongyong*, when explaining the sentence "What is given to the people by *tian* is called nature" (*tianming zhiwei xing* 天命之谓性), Kang Youwei said:

> Humans cannot be made by humans; humans are born by *tian*. Human nature is the stuff of human life. Humans derive their nature by spiritizing *qi* which

is received from *tian*, not by obtaining physical body from their parents. (Kang 2007, vol. 5, p. 369)

In *Datongshu*, he further explained that:

Humans are born by *tian*. It is by the bodies of one's parents that humans come to the world. However, this does not mean parents can dominate their children. Because every individual ultimately belongs to *tian* and not to their parents. (Kang 2007, vol. 7, p. 36)

According to these statements, "humans are born by their parents" only describes the formation of the human body in the world of experience. However, a person cannot be called a "human" simply by virtue of their bodies. To illustrate this, Kang Youwei gave an example: "When I lived in village as a child, I used to see a mad man. His mother and wife fed him but he didn't eat. He just put his fingers into his mouth and bit them. A man such as this only has a human body but no spirit. This is why he could hardly be called a human." (Kang 2007, vol. 5, p. 561). In other words, it is human nature rather than the visible figure or body that makes us human. The phrase "humans are born by their parents" can explain where the human body comes from, but it cannot metaphysically explain where human nature comes from. Therefore, "humans are born by *tian*" can be understood as "human nature is derived from *tian*".

However, in which way do humans get their nature from *tian*? Kang Youwei explained the process of the derivation of human nature from *tian* thus:

The infinite and flowing *yuanqi* 元气 ("primal *qi*", also variously translated into primal ether or primal force or vital force) created the heavenly and the earthly (*tiandi* 天地). *Tian* and humanity were all made of the same *yuanqi*. Though one is great and one is tiny, they both share the same *yuanqi* that proceeded from *taiyuan*太元 (the great origin of the universe). The relation of humanity and *tian* is like when a drop of water is thrown into the ocean: not the slightest difference between them can be found. (Kang 2007, vol. 7, p. 4)

According to this argument, the natures of human beings and every other one of the ten thousand things are made of *qi*. It is the origin of the universe as well as the vital force that invigorates all things. "All the things in the cosmos originated from *yuanqi*. Humanity is only one of the creatures made of it" (Kang 2007, vol. 7, p. 49). Human nature is the luminous numinosity obtained through the process of gaining *qi*. As such, since humans are things, they have a fundamental *qi* constitution. In terms of the real world, a person is thrown into a world of inequality the moment they are born. People might be pretty or ugly in appearance, healthy or disabled in body; they might be born into a rich family or a poor one, a noble or base one. The preposition that "humans are born by *tian*" not only gave a new definition of humanity, but also provided a premise for equality in the *datong* society. From an a priori perspective, because every person was created by *qi* without any differences between them, human nature is realized via the principle of *datong* where "everyone is unified in their equality".

However, according to our common experience, humans are given birth to by their parents. It is because of this fact that *qinqin* 亲亲 (being affectionate to one's parents) is regarded as the most essential ethical principle of Confucianism. Does "humans are born by *tian*" contradict this aspect of traditional Confucianism? In fact, the proposition that "humans are born by *tian*" was not first proposed by Kang Youwei. Instead, it can be traced back to Dong Zhongshu 董仲舒 (BC179–BC104), a Confucian master from the Han dynasty. In his book *Chunqiu Fanlu* 春秋繁露 (Luxuriant Gems of Spring and Autumn Annals), Dong Zhongshu said: "Those giving birth to humans cannot create humans, humans are created by *tian*. *Tian* is like humans' great grandfather" (Su 2002, p. 318). By saying that "Humans are created by *tian*", Dong Zhongshu actually meant that humans derived their nature from *tian*. In the chapter "Shencha Minghao 深察名号", he explained the meaning of *xing* 性 (usually translated as nature or human nature): "Nowadays people are confused by the meaning of *xing*. There are various explanations of it. Why do we not investigate

the word *xing* itself? *Xing's* original meaning is *sheng* 生 (meaning to live, life or to give birth to, to generate). A thing's nature is that natural resource which it draws on for living. *Xing* is the essence of things" (Su 2002, p. 291). Dong Zhongshu asserted that *xing* and *sheng* had a close semantic relationship. *Xing* referred to the character of humans or other things which was naturally endowed at birth. It was not endowed by those who gave birth, but endowed by *tian*. Nonetheless, Dong Zhongshu also admitted the fact that humans were biological products of their parents. In the chapter "Shunming 順命", he said: "The father is like *tian* to his son; and *tian* is *tian* to the father. Nothing can be born without *tian*. *Tian* is the ancestor of all living things (Su 2002, p. 410). In the book *Chunqiu dongshixue* 春秋董氏学 (On Dong's Study of the Spring and Autumn Annals), Kang Youwei further elaborated Dong Zhongshu's theory: "Human inner dispositions and outer conditions alongside their capacities to know and perceive originate in *tian* and the shape of the human body originates in their ancestors". (Kang 2007, vol. 2, p. 375). According to Kang Youwei, the birth of humans could be understood in two dimensions: one was being born as the children of parents, the other was being born as the children of *tian*. Therefore, "Humans are born by *tian*" does not contradict the fact that humans are biological products of their parents.

Despite claiming that his theory was developed from Dong Zhongshu, Kang Youwei had to explain why Confucius highly praised the value of *qinqin* in most of the other classics. His answer was that "Confucius had preset the Law of the Three Ages and hoped to realize *datong* in the future". (Kang 2007, vol. 5, p. 379). "The Law of the Three Ages" was Kang Youwei's development of the theory of the Three Ages which first appeared in the Confucian classic *Chunqiu gonngyangzhuan* 春秋公羊传 (Gongyang Commentary on the Spring and Autumn Annals). It referred to the Age of Disorder, the Age of Approaching Peace, and the Age of Great Peace (also referred to as the Age of *Datong*). Kang Youwei applied this theory in explaining the evolution of human civilization. According to him, each Age resembled a stage of civilizational development. The Age of Disorder was the primary stage of human civilization where people lived in savagery and needed to be cultivated and regulated by hierarchical rules and regulations. The Age of Approaching Peace was an advanced stage of human civilization where laws and regulations tended to be more equal to the people. The Age of the Great Peace was the final stage of human civilization where all hierarchical systems would disappear, and distinction and boundaries between people would be eliminated. Each Age had its corresponding systems, laws, and values. These systems, laws, and values might contradict each other due the different civilizational stage they belonged to. Kang Youwei asserted that the order of the Three Ages could not be reversed. Although *datong* was the best prescription to save the world, "even Confucius himself could not apply the way of Great Harmony to his time, since he lived in the Age of Disorder. He had to obey the order of the Three Ages". Therefore, Confucius' advocation for the hierarchical order was his solution to the Age of Disorder. The hierarchical regulation and rituals of the classical Confucians was a necessary step on the way to *datong*.

By the phrase "eliminating boundaries and distinctions", Kang Youwei did not mean to take them down by force, but instead to dissolve them by *ren* 仁 (often translated as benevolence, humanity, or consummate conduct). *Ren* is the core term of traditional Confucian ethics, yet Kang Youwei understood it in a novel way:

> The heart which cannot bear to see the sufferings of others (*buren ren zhi xin*不忍人之心) is called *ren*, or electricity, or ether. All the men have it, therefore we can say the human nature is good. (Kang 2007, vol. 5, p. 414)

> The thing that makes humans superior to any other creature is their nature because they have a natural disposition to virtue. It is like metal being attracted by a magnet. It is because humans have this ether inside them that they can be attracted by virtue. (Kang 2007, vol. 5, p. 426)

Traditional Confucianism took *ren* as an abstract virtue. However, Kang Youwei understood *ren* as having some kind of physical attribute of *qi* which was naturally a part of human nature. It was by this physical attribute that humans could be affectionate to each other and be inclined toward kindness. Kang Youwei explained the meaning of *ren* thus: "*Ren* is expansive love" (*boai* 博爱) ([Kang 2007](#), vol. 6, p. 424). He further explained: "I am endowed with *qi* derived from *tian*. All the living things are all endowed with *qi* derive from *tian*. Therefore, they are all my brothers and sisters. How can I not love them?" ([Kang 2007](#), vol. 5, p. 415). Because humans shared the same *qi* with the ten thousand things, they are all equally relevant to us and we are responsible for taking care of them. The complete realization of *ren* is to love all the other living creatures with impartiality.

Therefore, according to Kang Youwei, the development of the Three Ages is accompanied by the expanding and spreading of *ren*. He explained the relation between the two processes by quoting Mengzi's theory of *qinqin* (to treat one's parents affectionately), *renmin* 仁民 (to treat the people humanely), and *aiwu* 爱物 (to love all the living creatures):

> The following was the law of the Three Ages set up by Confucius: During the Age of Disorder, *ren* could not be spread broadly so people were only required to be affectionate to their parents. During the Age of Approaching Peace, *ren* could be spread within the same species so people could be kind to each other. During the Age of the Great Peace, all living creatures are equal so people could love all creatures. If there are differences in how one employs *ren*, then there will be progress and retrogression, largeness and pettiness in the world. ([Kang 2007](#), vol. 5, p. 415)

From *qinqin* to *renmin* to *aiwu*, all the distinctions and differences between people, classes, nations, and species gradually disappear with the development of the scope and range of *ren*. In short, in order to make his *datong* theory more convincing, Kang Youwei built up a system including the theory of human nature and the theory of Three Ages that he claimed were all based on core concepts in the Confucian tradition.

However, many of his arguments remain questionable. First of all, his interpretation of Mengzi's theory is suspicious. According to Mengzi, the extension of *ren* followed the sequence from being affectionate to parents to being humane to the people and finally to loving all creatures. However, the affection for parents, people and all living creatures is different at each stage. Mengzi's exact words were "In regard to inferior creatures, the superior man is kind to them, but not loving. In regard to people generally, he is loving to them, but not affectionate. He is affectionate to his parents, and lovingly disposed to people generally. He is lovingly disposed to people generally, and kind to creatures." ([Jiao 1987](#), pp. 948–49). It is clear that for Mengzi, the further *ren* extends—from family to the people to all living things—the more diluted the "love" for the object. Therefore, Mengzi used the different phrases of "to be affectionate", "to be kind", and "to love" to describe the different relationships between one person and a relational object. It is a natural response that humans have different feelings and affections when facing different objects. It was based on this natural fact that the Confucian differential arrangement of relationships was established. If there is no difference between the affection for parents, the people, and all living things as Kang Youwei claimed, then the affection for parents cannot be the foundation for the affection in the other relationships because the affection for the people will dissolve the affection for parents and the affection for living creatures will further dissolve the affection for the people.

Secondly, according to Kang Youwei, the *datong* society was formed by millions of completely self-sufficient independent and equal individuals (what he called duren 独人 or "solitary men"). If this were the case, there would be no need for people to socialize with others. As such, it is hard to imagine these "solitary men" could keep close attachment to each other as Kang Youwei asserted.

Moreover, even though Kang Youwei maintained that "there was a fixed track" for the progression of the Three Ages "that could not be surpassed", and even though he did not want his *datong* theory published before his death, because the ultimate goal of humanity

was on the horizon, he wondered what possible forces could thwart the people's advance toward the *datong* society.

### 4. Preserving Social Relations: Chen Huanzhang's Reflection on Kang Youwei's Theory

Being the most important thought of Kang Youwei, all of his students were deeply familiar with his *datong* theory. However, in 1922, one of his favorite students, Chen Huanzhang 陈焕章 (1880–1933), published "Cunlun pian" 存伦篇 (Article of Preserving Social Relations) proposing his own understanding of *datong* that greatly differed from his teacher's.

Chen Huanzhang was born in the town of Gaoyao, Guangdong province. He began studying with Kang Youwei at the Wanmu Academy at fifteen years old and obtained the title of *jinshi* 进士 (a successful candidate in the highest imperial examinations) in the year of 1904. One year later, he was selected by the Qing government to study in America. He obtained his PhD from the University of Columbia with a dissertation titled *The Economic Principles of Confucius and his School* 孔门理财学. At Kang Youwei's behest, Chen Huanzhang established the Shanghai Confucius Association after returning to China in 1912. He also started several journals to advocate the doctrine of Confucius and expand the Confucius Association worldwide[5]. According to Chen Huanzhang's own account, he became suspicious of Kang Youwei's *datong* theory early on: "I used to suspect the theory of *datong* twenty years ago. In my dissertation *The Economic Principles of Confucius and His School* and in the article 'On Confucianism' written in Shanghai several years ago, I made a little argument on the theory. I had also discussed the theory with many of my Chinese and foreign friends over the years but did not draw my own conclusion until the year 1915. When writing the book *Administering State Affairs Under the Instructions of Confucius* 孔教经世法, I pondered the Liyun' chapter over and over, and by studying the *Spring and Autumn Annals*, I came to realize the five social relations must exist in the time of *datong* described in the 'Liyun.' Thus my confusion lasting for more than ten years had come to a resolution." (Chen 2015, p. 79).

Of the five social relations, Chen Huanzhang cared most about the first three ones, that is, the relations between ruler and ministers, father and son, and husband and wife. He was concerned with these three because the former was a public relation while the latter two were domestic relations. Therefore, in order to demonstrate that the five social relations must all be preserved in the *datong*, his argument focused on the reason why these three are necessary features of the *datong* society.

First of all, in defense of the relation between ruler and ministers, Chen Huanzhang said:

Recently, there was a misunderstanding of *datong* in the "Liyun" that argued there were no rulers and ministers in the *datong* society. But this is wrong. In the *datong* society, the world was a community of people. This means that the sovereign did not pass the throne to his own son but selected "worthy and capable people" for government office. But this does not mean there were no ruler or ministers. If that were the case, then why did they need to "select worthy and capable people" to govern? It was because of the distinction between the ruler and ministers and the fact that not everyone was equal that the noble could rule the base and the capable could rule the incapable. Otherwise, what was the point of selecting the worthy and capable if they were not appointed? Since there were positions for the worthy and capable, it was obvious that they played a role as the rulers and superiors. This can be clearly seen in the text. Besides, the text says "people are reliable and seek harmony amongst each other." If there were no ruler and ministers the people would be like a pile of sand. Who would encourage them to be honest and seek harmony? Who would be responsible for their moral cultivation? If there were no one taking the responsibility, even though honesty and harmony were encouraged before, laws and rules would be fickle, and wars and fights would run rampant. Therefore, even *xiaokang* 小康 society could hardly be realized, not

to mention the *datong* society. In conclusion, the *datong* society must have a ruler and ministers. (Chen 2015, p. 95)

Compared with Kang Youwei, Chen Huanzhang returned to the text of the "Liyun" itself and based his explanation of *datong* thereon. He clearly recognized the authority of the classical text. It can be inferred from the text that *datong* was definitely not a society without ruler and ministers.

In addition, Chen Huanzhang gave a new connotation of the word *jun*君 (ruler, lord):

*Jun*, in terms of its pronunciation which sounds like *qun* 群 (group or crowd), means to group. If every individual in the world is independent and isolated and has no need to socialize with others, then there can be no ruler. However, if two people come together to form a group, there must be subordinates and rulers. For example, in the relation between husband and wife, the husband is the ruler. In the relation between father and son, the father is the ruler. In the relation between elder and younger brothers, the elder brother is the ruler. In the relation between two companions, the capable one will definitely be the ruler. This is the natural way of human society. (Chen 2015, p. 93)

Chen Huanzhang emphasized that *jun* did not necessarily refer to the sovereign but that the relation between ruler and subordinates exists in many occasions. Humans cannot live in the world in isolation. They need to communicate and socialize with one another, therefore, they must follow certain orders and rules. These kinds of relations can both find expression in the strict hierarchical relationship between emperor and ministers as well as the less hierarchical relationships that obtain between superiors and inferiors. Even under the public democratic system there must be someone playing the role of the president. The president in this sense can also be viewed as a form of *jun*. "A ruler is necessary if the state and world are to be governed. It does not matter whether that ruler is an emperor or a president" (Chen 2015, p. 96).

Apparently, Chen Huanzhang tried to respond to Kang Youwei's criticism on the relation of ruler and ministers. According to *Datongshu*, the relation between ruler and ministers was merely a kind of class oppression. The royal families made themselves superior to the common people, thus causing the "suffering of class oppression". Chen Huanzhang realized that Kang Youwei had overstated the conflict between ruler and ministers without considering the necessity and rationality of hierarchical order in human society. Kang Youwei's understanding of the relation between ruler and ministers was based on his theory of human nature. Because "humans were born by *tian*", any distinction and difference between humans was not inherent, but rather was something acquired. Yet in the perspective of Chen Huanzhang, the existence of *jun* accorded with human nature. He said: "To group is the nature of humans. Thus, the existence of *jun* accords with human nature" (Chen 2015, p. 96). The social differences and distinctions found in human society are natural and justified; they are the foundation of political order in human society. Even Confucius admitted the people's talents were different, some were born "who already knew" and some were born who had to "learn to know".

Next, in the defense of the relation between father and son, Chen Huanzhang realized Kang Youwei's understanding of *qinqin*, *renmin*, and *aiwu* was very questionable. To illustrate the function of *qinqin* and *renmin*, Kang Youwei once used a metaphor of "abandoning the boat to step ashore." He only took the *qinqin* and *renmin* as tools and methods to achieve *datong*. When *datong* was realized, *qinqin* and *renmin* were to be abandoned. However, In Chen Huanzhang's opinion, the development of human affection is like planting or building. Without *qinqin*, the affection for others was like water without a source, or a tree without a root, something quite unimaginable:

"People did not only love their own parents but also loved the parents of others. They cared not only their own children but also for the children of others". By the use of "parents" and "children", it is apparent that *datong* started with being affectionate to family members. How could this contradict "the world is that

of the community"? Otherwise, although one claims to love the others' parents and to care for the others' children, this love is like water without a source, a tree without a root. These are things that are entirely impossible. (Chen 2016, p. 20)

From the different words used in "Liyun" such as "their own parents", "parents of others", "their own children" and "children of others", it can be concluded that the love for one's own parents and children was different from the love for others' parents and children. Therefore, "People did not only love their own parents but also loved the parents of others . . . .They did not only care for their own children but also cared for the children of others" means that it is only after people render good care for their own parents and children that they then can give consideration to the parents and children of other people. In fact, the sentence "people did not love their own parents" has proven to be particularly problematic for many Confucian commentators throughout history. According to the Tang dynasty Confucian scholar Kong Yingda 孔穎達, this sentence meant that "The sovereign was unselfish, he spoke with honesty and behaved with kindness so the people imitated him. Therefore, they did not love their own parents, and did not care for their own children". Furthermore, he said that "they make it so that the elderly have what they need to live out their lives" means since the whole world was united, people were not only affectionate to their own family members but supported all the elderly in the world regardless of kin relations (Zheng and Kong 2008, p. 878).

By saying "The whole world became one unity and people were not only affectionate to their own family members", Kong Yingda seemed to imply a love without differences. But the establishment of an ethical order according to social relations was the basic principle of Confucianism. Therefore, Zhang Zai 張載 (1020–1077), a Confucian master from the Song dynasty, argued by saying that "people loved their own parents" did not contradict the proposition that "people did not only love their own parents" because it only represented the period of "lacking and narrow compassion." When it came to the period of "unobstructed love", people would not only love their own parents but also the parents of others (Wei 1985, p. 253). Of course, the unspoken consensus was that the affection for people's own parents and for the parents of others was different in degree and in form.

Chen Huanzhang knew that Kang Youwei's understanding of the love that paid no heed to different social relations in the *datong* society was based on his theory of human nature. He simply did not agree with it. He emphasized the decisive role of parents in the birth of humans: "The birth of humans must have its origin. It is the parents who give birth, and those who are born are called children. This fact cannot be denied under any circumstances. If there is a man, he could not come down from heaven nor could he grow up from the earth, instead, he must be given birth to by his parents. So when treating his parents he must follow the rules of social relations" (Chen 2015, p. 85). Kang Youwei's theory of human nature did not follow the observation of the world of experience, but was instead based on a theoretical deduction whereas Chen Huanzhang returned to the common-sense experience and drew the conclusion that "the love between father and son is rooted in nature, their binds are tight and cannot be broken" (Chen 2015, p. 88).

Furthermore, Chen Huanzhang tried to prove that there must be marital relations between husband and wife in the *datong* society:

Confucius discussed the *datong* system saying that "all the women have a place to return to." He obviously referred to the families of their husbands and their parents. Since women need to get married to obtain their social role, therefore, families mainly refer to the family of their husband's [sic]. Zheng Xuan explained "all the women have a place to return to" as "they all married into good families." If free love took the place of marriage, what would happen if a man without good virtue abandoned his spouse? She would have no husband's family to return to. Her brothers would not know what she suffered and would laugh at her so that she could not return to her parents either. Confucius said "all the women have a place to return to" instead of saying that "all women have their families",

because only by marrying into her husband's family could a woman settle down and achieve real independence and liberty. (Chen 2016, p. 21)

When explaining the "Liyun" text, Kang Youwei deliberately changed the word "to return" (*gui* 归) to "to tower over" (*kui* 归). Because in his point of view, the word "to return" itself showed the dependence of women on their husbands: "What does 'women cannot set up their own families' mean? A woman joining with a man is called marriage or 'returning.' This is where the principle of 'husband is the guide of wife' originated. That 'women should listen to husbands in marriage' was considered the highest virtue. However, women had lost their independence and rights. This seriously violated the principle of equality!" (Kang 2007, vol. 7, p. 57). By changing the word "to return" to "to tower over", he claimed the independence of women: "'To tower over' means towering majestically like a mountain. Even though women are weak, they can achieve majestic independence and avoid oppression. The husband and wife should make a contract and adhere to it. This is the principle of husband and wife". (Kang 2007, vol. 5, p. 555). But Chen Huanzhang accepted the traditional explanation which translated "to return" as "to marry" (*jia* 嫁). As for women, they do not have a family of their own. Before marrying, "family" refers to the family of their parents and after marrying "family" refers to the family formed with their husband. As such, the phrase "all the women have a place to return to" means the existence of marriage in the *datong* society.

In addition, Chen Huanzhang pointed out another flaw in Kang Youwei's theory of human nature:

Since *tian* did not make men and women physically the same, therefore marriage cannot be abandoned to be replaced by free love. (Chen 2016, p. 21)

In fact, despite claiming that all humans were born by the same *qi* substance, Kang Youwei also admitted that people were born with certain differences in the real world. He borrowed the Han Confucian theories on *yin* and *yang* that were used to explain the origin of human nature and emotion. He said the *qi* of *yang* made the spirit of humans and the *qi* of *yin* formed the figure of humans. The birth of humans in the world of experience was the combination of *yin* and *yang*. Though the *qi* of *yang* was clear and full of benevolence, "When a person comes into the world they are inevitably influenced by the *qi* of *yin* which constitutes their bodies" (Kang 2007, vol. 5, pp. 426–27). Chen Huanzhang noticed that no matter how equal the talents, abilities, and social status of men and women could be, they still had biological differences. This difference would ultimately lead to certain relations between men and women. His opinion can be supported by the traditional understanding of the relation between husband and wife in Confucianism. In the "Hunyi 昏义" (The Meaning of Marriage) chapter in the *Liji*, we read, "When the distinction between the male and the female is formed then the moral integrity between the husband and wife is established". (Zheng and Kong 2008, p. 2277). According to Confucianism, the relation of husband and wife was naturally established on the physical distinction between men and women.

Another criticism on marriage made by Kang Youwei was that women were confined to a life of dependence in marriage. If a woman is kept in a terrible marriage and has no way to escape, she would have no choice but to endure it the rest of her life. Hence, Kang Youwei invented a new way for men and women to get along with each other in the *datong* society: they could make a contract to maintain their relationship. The couple could also determine the length of the contract themselves. In addition, while the contract is in effect, if either side changed their mind, he or she could freely withdraw from it. And when the contract expires, they could choose to renew it or not. But Chen Huanzhang tried to demonstrate that marriage was not the suppression of women. He took pregnancy as an example saying that "Only women can get pregnant, men cannot. [If free love is permitted, then] during the period of pregnancy, women cannot have relations with other men, however men can still have relations with many other women. This is not fair to pregnant women". (Chen 2015, p. 83). Again, Chen Huanzhang focused on the biological differences of men and women to illustrate the fairness of marriage. The fact that only females can get pregnant

determines that the male and female cannot have the same freedom in the relationship. In light of this, he asserted the positive side of traditional marriage in preserving women's independence and liberty:

> According to our Chinese customs, women are in charge of the affairs of the household and men are in charge of the affairs of the public. There is a clear division in the work that men and women do. There was no so-called "inequality" between husband and wife. Besides, after women have children, they not only enjoy the rights of women but also the rights of mothers. In the *Spring and Autumn Annals*, even the emperor of the Zhou dynasty had to bear the criticism of not serving his mother. (Chen 2015, p. 84)

The traditional Chinese family pattern encouraged women to show their talents and abilities in domestic affairs and had priority over men who were responsible for public affairs. In other words, they enjoyed considerable freedom and priority within the family and could get more protection and security by marriage.

Furthermore, Kang Youwei assumed that all humans wish to seek happiness and avoid suffering. Everything he designed for the *datong* society embodied this principle. But what is suffering? What is happiness? Happiness is generally understood as the satisfaction of the desires for good food, beautiful sights and sounds, safe and comfortable living conditions, and good health. But Chen Huanzhang saw the complexity of human nature and that desires for happiness varies. Not all the happiness referred to sensual pleasure. In Kang Youwei's designs for the *datong* society, the most appealing regulation was that he removed the restriction of marriage on men and women. They could freely combine and separate with each other by following their hearts. In Kang Youwei's mind, this was the only way that human nature could find realization. However, Chen Huanzhang pointed out another side of human nature, that is "If people only listen to their natural desires to combine as couple or separate like strangers then affection for each other becomes irregular and fragile, just like water flowing off the back of duck or wind blowing through duckweed. How can that be happiness? This seriously violates human nature." (Chen 2015, p. 82). Besides desire and passion, humans are also eager for stable emotional bonds and relationships. Marriage and family are the systems that protect these kinds of emotions and affections. Indulgence in lust and passion does not necessarily bring human happiness; it can contrarily also ruin the wonderful experience of having stable and lasting affection for others.

Thus, it is clear that underlying the two notions of "eliminating social distinction" and "preserving social relations" were Kang Youwei's and Chen Huanzhang's two different understandings of human nature. Kang Youwei deliberately avoided describing the human from an empirical perspective and instead set up his theoretical system on the preposition that "humans are born by *tian*." However, Chen Huanzhang returned to the traditional understanding of human nature; therefore the social relations were the accomplishment and safeguard of human nature.

### 5. "The Whole State Went Mad" and the Significance of Two Understandings of *Datong*

Why did Chen Huanzhang return to Kang Youwei's *datong* theory nearly twenty years after he first proposed it? Is there any other reason besides that he felt Kang's theory was questionable? The answer is yes. With the Republic of China replacing the old empire after the 1911 revolution, Kang Youwei felt that he was living in a time that corresponded with the final *datong* stage in his theory of the Three Ages. However, Chen Huanzhang was much less concerned with an ideal that could be and focused more on the actual social conditions of his time:

> Nowadays the whole state has gone mad and morality has decayed. Social relations are undefined and, to make things worse, there are those who suggest that the relation between ruler and ministers can be abandoned because China has become a republic, that the relation between husband and wife can be abandoned

because promiscuity was prevailing, and that the relation between father and son could be abandoned because of the "family revolution." Since the "three cardinal guides" had been abandoned, all human affairs have been put out of order and evil speech and violence abound. This is an unprecedented disaster never before seen in five thousand years. (Chen 2015, p. 82)

Even though the establishment of the new republic saw an end to political revolution, that did not mean that there was an end to ideological revolutions. Various doctrines and ideologies from both the Eastern and Western worlds flooded into modern China, strongly impacting the traditional lifestyle and thoughts. Among them, one of the mainstream ideologies was individualism and liberalism. The New Culture movement started in 1915 strongly criticized the feudal family system as binding the individual. For example, Chen Duxiu 陈独秀 (1879–1942) wrote an article claiming that "loyalty, filial piety, chastity, and righteousness" were the "morality of slaves." (Chen 1915, vol. 1, no. 1). Chen Huanzhang was very sensitive to the problem of the dissolution of social relations for two reasons. On the one hand, as he mentioned many times in his "Article of Preserving Social Relations", without the relation between father and son, parents would be unwilling to give birth to children. Without the relation between husband and wife, there would be abuse of contraception and abortion. He called them "the way to exterminate humans." Family must be protected out of the consideration for the survival of humanity. On the other hand, "Today people desire to spread far and wide and all kinds of strange and bad things emerge. These are people who ride the waves wherever they go. Others pretend to uphold the Confucian classics and adorn their new thinking with its accoutrements, daily paying lip service to the *datong* society, but in doing so they completely misunderstand it" (Chen 2015, p. 79). The *datong* theory of Kang Youwei was used by those shallow people as a weapon to attack family and social relations. For Chen Huanzhang, who was dedicated to advocating Confucianism, if the doctrine of Confucianism could not be adhered to, then it would finally be replaced by another doctrine and ideology.

## 6. Conclusions

Kang Youwei's *datong* theory was apparently not the simple and superficial utopian imagination that it was generally considered to be. In fact, it involved the discovery of many hidden traditions in Confucianism. Regardless of whether it was the "Liyun" chapter or the theory of human nature offered by Dong Zhongshu, they all contained content that greatly challenged the mainstream of Confucianism, which is worth attention. In recent years, there are scholars suspecting that traditional *wulun* pattern in Confucianism could hardly respond to the problem of interactions with strangers (See Zhao 2007, pp. 15–21). Kang Youwei offered a possible way to approach the problem. However, it has to be pointed out that Kang's *datong* theory should be considered as an attempt rather than a solution to the problem. Being one of the few who could really see the flaws in Kang Youwei's theory, Chen Huanzhang's vigilance and attempt to prevent the dissolution of traditional Chinese social relations in the early republican period of modern China provide us with a good mirror by which to reflect on and to further the study of Kang Youwei's theory of *datong*.

**Funding:** This research was funded by Youth Project of The National Social Science Fund of China, grant number 20CZX019 and the article processing charge was funded by Chinese government financial allocation.

**Informed Consent Statement:** Not applicable.

**Data Availability Statement:** Not applicable.

**Conflicts of Interest:** The author declares no conflict of interest.

## Notes

[1] Federico Brusadelli called it a "universal pattern" (Brusadelli 2020, p. 56).

[2] Scholars including Fan Wenlan, Li Zehou and Tang Zhijun all agreed that Kang's *datong* theory was a theory independent from Confucianism invented by himself, aiming at opposing feudal autocracy and leading the way for the modern bourgeoisie. Fuller discussions see Fan (1955); Li (1979) and Tang (1984). Zhu Weizheng asserted that Kang Youwei's *datong* theory was just one of the western utopian theories flooded into modern China with little value. Its academic value is "to expose and criticize the problem of Chinese society". See (Zhu 1997, p. 247). Goran Malmqvist also agreed that the *Datongshu* which shared some creeds with the Communist Manifesto might have been influenced by utopian novels such as Looking Backward written by Edward Bellamy. But he claimed that Kang established the utopia imagination on his belief of Confucianism. See Malmqvist (1991).

[3] For example, Wang Hui claimed that Kang Youwei's *datong* imagination was the vision combining the universalism of Confucianism and the western science, political and religious knowledge. "(Wang 2008, p. 826). Besides, Takeuchi Hiroyuki noticed that Kang Youwei's *datong* theory had inherited the legacy of many other thinkers such as Wang Tao, Chen Qiu in earlier time. See Hiroyuki (2008). A recent study on Kang Youwei's *datong* theory was made by Federico Brusadelli. He said, "beyond any doubt that the Datong Shu cannot be considered as a unique specimen in Kang's production, nor as a fancy appendix to it. It was not a final detour from a more rational political path, nor the culmination of a "second phase" of his thought. Instead, it must be considered as fully embedded in Kang's earliest reflections on the meaning of tradition and on the trajectory of human history and mundane institutions." (Brusadelli 2020, pp. 40–41).

[4] As many scholars have pointed out, Kang's *datong* theory inherit many of the Huayan and Mahayama tradition (Brusadelli 2020, pp. 44–50). However, this does not mean his *datong* theory does not obey Confucianism. Specific argument can be seen below.

[5] Chen Huanzhang's efforts of establishing Confucian Society and his view of Confucian religion has been receiving much attention in recent years. See Gan Chunsong: Kang Youwei, Chen Huanzhang and the Confucian Society, in *Contemporary Chinese Thought*, vol. 44, no. 2, pp. 16–38.

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
