# Peer review of "“Eliminating Social Distinctions” or “Preserving Social Relations”: Two Explanations of Datong in Modern China"

_religions, doi:10.3390/rel13080720_

Round 1

Author Response

I am greatly appreciative for this reviewer’s comments and for pointing out relevant issues with the article.

Point 1:  The argument should be made clearer. The first section only provides a general introduction to Chinese social relations, Kang Youwei’s identity as a Confucian, and his Datong shu. It does not mention what this article tries to argue. Also, the article lacks a conclusion.

Response 1: I add a paragraph in the first section to illustrate that this article tries to argue that the new social relation pattern revealed by Kang Youwei’s datong theory which was different from the traditional wulun pattern should also be viewed as a “Confucian” pattern. Also I add another paragraph in the end of the article to give a conclusion.

Point 2: The author does not refer to any English publication. Actually, he/she consulted very few scholarly works. There is a Chinese article published in Zhongguo zhexueshi 中國哲學史 (History of Chinese Philosophy), issue 3 (2020), which also compares Kang Youwei’s and Chen Huanzhang’s ideas about datong, and its content is very similar to the current article. This article should be mentioned and the difference between it and the current article should be pointed out.

Response 2: In the revised manuscript, I consult more scholarly works (both Chinese and English) which can be seen in the footnotes and reference. Since the article published in Zhongguo zhexueshi 中國哲學史 was written by the same author of this article, therefore, they do share many similarities in ideas and resources. However, the two articles have different focuses. The article published in Zhongguo zhexueshi mainly discussed the Chen Huanzhang’s reflection on Kang Youwei’s datong theory. Chen is the focus. However, in this article, I try to demonstrate Kang’s datong theory obeyed and developed many hidden traditions in Confucianism. Kang is the focus.

Point 3:The significance of the comparison between Kang Youwei and Chen

Huanzhang should be highlighted and elaborated.

Response 3: The significance of the comparison between the two thinkers has been elaborated in the beginning and the conclusion part.

Point 4: The format of references/footnotes should be made consistent.

Response 4: I’ve modified the reference/footnotes to make them more consistent.

Thank you very much

Reviewer 2 Report

This is a very interesting subject and the way you have tackled it is very good and interesting. You obviously know the Chinese-language and primary sources very well indeed and have put forward a coherent and interesting idea about the nature of Confucian thought and Kang Youwei's contribution to it and to Chinese philosophy and ideas in general. Overall, you have definitely contributed to the literature.

Here are four ways I think the article could be improved.

1. There needs to be a specific conclusion which sums up what you have contributed to the field you are working in and why your contribution matters. The final section about Chen Huanzhang is of course interesting and relevant, but I think a conclusion is still necessary, and your article loses a great deal without it.

2. A bit of reference to non-Chinese material would make your article even better than it is at present. I'm fully aware the most important research in a topic like this is done in Chinese, but there is some work in European languages, especially English, that is well worth looking at. You need to engage with the literature on the topic, and if you engage only with what's in the Chinese language you do miss a perspective that could be useful. If you look at it and find it is hopeless that is your prerogative, but I think you should at least look at it. Even your engagement with Chinese-language materials seems a bit limited to me.

3. In some places the English-language expression is not specially good, and I noted a few outright mistakes. Here are a couple of specific instances to show what I mean.

On p. 5, line 214, "originary qi" as a translation of yuanqi is not a good usage, as the word "originary" is not in normal use. On p. 7, line 356, the word "unspeakable" does not fit the context and is very unclear. "Unspeakable" usually means something is so bad, you just can't mention it, but the context suggests that's not quite what you mean here. On p. 9, lines 462-3, the sentence "They did not only care their own children but also cared for the children of others" should read "They cared not only for their own children but for also for the children of others". As it stands, it is not clear and not good English.

4. Although the romanization is generally excellent, I did notice a couple of errors, perhaps more typing errors than outright mistakes. For example, on p. 13, line 677, Kongjia should be Kongjiao.

Author Response

I am greatly appreciative for this reviewer’s comments and for pointing out relevant issues with the article. 

Point 1. There needs to be a specific conclusion which sums up what you have contributed to the field you are working in and why your contribution matters. The final section about Chen Huanzhang is of course interesting and relevant, but I think a conclusion is still necessary, and your article loses a great deal without it.

Response 1:  I add a paragraph in the end of the article to give a conclusion to illustrate the main purpose of the argument the the significant of comparing Kang and Chen’s datong theories.

  1. A bit of reference to non-Chinese material would make your article even better than it is at present. I'm fully aware the most important research in a topic like this is done in Chinese, but there is some work in European languages, especially English, that is well worth looking at. You need to engage with the literature on the topic, and if you engage only with what's in the Chinese language you do miss a perspective that could be useful. If you look at it and find it is hopeless that is your prerogative, but I think you should at least look at it. Even your engagement with Chinese-language materials seems a bit limited to me.

Response 2: In the revised edition, I consult more scholarly works (both in Chinese and English) which can be seen in the footnotes and reference.

Point 3&4. In some places the English-language expression is not specially good, and I noted a few outright mistakes. Here are a couple of specific instances to show what I mean.

On p. 5, line 214, "originary qi" as a translation of yuanqi is not a good usage, as the word "originary" is not in normal use. On p. 7, line 356, the word "unspeakable" does not fit the context and is very unclear. "Unspeakable" usually means something is so bad, you just can't mention it, but the context suggests that's not quite what you mean here. On p. 9, lines 462-3, the sentence "They did not only care their own children but also cared for the children of others" should read "They cared not only for their own children but for also for the children of others". As it stands, it is not clear and not good English.Although the romanization is generally excellent, I did notice a couple of errors, perhaps more typing errors than outright mistakes. For example, on p. 13, line 677, Kongjia should be Kongjiao.

Response 3&4: I’ve corrected the typo errors and modified the English in the manuscript.

Reviewer 3 Report

It is a very good paper, but it just touches on a very few secondary sources in the opening section about Kang Youwei and Chen Huanzhong. The author could do a better job of presenting how they are generally regarded by contemporary scholars, even a more expansive set of footnotes citing this work would allow the author to situate his paper within the current state of studies on Kang.  Still, the paper as a whole does a great job of presenting and interpreting the work and ideas of Kang and Chen.

There are too many long paragraphs, so the author should break them down. The paper also needs a better English editing, particularly in the opening pages.

Author Response

I am greatly appreciative for this reviewer’s comments and for pointing out relevant issues with the article. 

In the revised edition, I consult more scholarly works (both in Chinese and English) which can be seen in the footnotes and reference.

I’ve modified the reference/footnotes to make them more consistent.